# Developing Sustainable Careers during a Pandemic: The Role of Psychological Capital and Career Adaptability

**Jetmir Zyberaj** [1,*] , **Sebastian Seibel** [1,2] , **Annika F. Schowalter** [1] , **Lennart Pötz** [1] ,
**Stefanie Richter-Killenberg** [1,3] **and Judith Volmer** [1]

[1] Work and Organizational Psychology Group, Department of Psychology, University of Bamberg,
96047 Bamberg, Germany; sebastian.seibel@uni-wuerzburg.de (S.S.);
annika.schowalter@uni-bamberg.de (A.F.S.); lennart.poetz@uni-bamberg.de (L.P.);
stefanie.richter-killenberg@uni-bamberg.de (S.R.-K.); judith.volmer@uni-bamberg.de (J.V.)

[2] Work and Organizational Psychology Group, Department of Psychology, Julius Maximilian University of
Würzburg, 97070 Würzburg, Germany

[3] Social Psychology Group, Faculty of Psychology, Technische Universität Dresden, 01069 Dresden, Germany

\* Correspondence: jetmir.zyberaj@uni-bamberg.de

**Abstract:** The Coronavirus disease 2019 (COVID-19) has not only had negative effects on employees' health, but also on their prospects to gain and maintain employment. Using a longitudinal research design with two measurement points, we investigated the ramifications of various psychological and organizational resources on employees' careers during the COVID-19 pandemic. Specifically, in a sample of German employees ($N = 305$), we investigated the role of psychological capital (PsyCap) for four career-related outcomes: career satisfaction, career engagement, coping with changes in career due to COVID-19, and career-related COVID-19 worries. We also employed leader–member exchange (LMX) as a moderator and career adaptability as a mediating variable in these relationships. Results from path analyses revealed a positive association between PsyCap and career satisfaction and career coping. Furthermore, PsyCap was indirectly related to career engagement through career adaptability. However, moderation analysis showed no moderating role of LMX on the link between PsyCap and career adaptability. Our study contributes to the systematic research concerning the role of psychological and organizational resources for employees' careers and well-being, especially for crisis contexts.

**Keywords:** psychological capital; career adaptability; LMX; career engagement; career satisfaction; career coping; COVID-19

## 1. Introduction

The Coronavirus disease 2019 (COVID-19), declared a global pandemic crisis on 11 March 2020 by the World Health Organization (WHO), has caused major social, economic, and health ramifications for societies worldwide [1]. Many organizations went bankrupt, causing employees to struggle with their careers by facing job loss or entering into stressful work practices such as short-time work schemes or layoffs [2–4]. While societies struggle with the disease, its effects continue to be present in workplaces. Hence, for many employees COVID-19 means adapting their career goals, switching jobs, or coping with changes brought by this pandemic [5]. As a result, employees' career management skills such as their employability (defined as the "ability to identify and realize career opportunities" [6], p. 23) may also change. These implications make it imperative to take stock and investigate personal and organizational resources that could help employees deal with their careers during pandemics.

In times of crises and volatile economies, employees' career success (i.e., one's work and psychological outcomes and achievements [7]) is the result of personal [8,9] as well as social and organizational resources [10,11]. These resources are posited to be significant for

employees' career-related outcomes such as employability. For example, staying updated and seeking adequate feedback about their environment helps employees to actively adapt to the changing work environments [6]. Similarly, personal resources such as optimism and self-efficacy facilitate employees' experiences with changing and uncertain environments [12]. Moreover, social resources such as social support and employees' relationship with their supervisors have been claimed to be important for employees in rapidly changing and uncertain economies [10].

The purpose of the current study is threefold. First, we investigate the role of psychological mechanisms of employees in coping with crises such as the COVID-19 pandemic. Factors related to personality characteristics, e.g., coping styles, have long been noted as essential for dealing with crises or for persevering during difficult career-related stages. Therefore, we investigate the role of employees' psychological capital (PsyCap) for their career-related changes brought on by the COVID-19 pandemic. PsyCap authors [13] noted that PsyCap consists of hope, efficacy, resilience, and optimism, which interact to help individuals with challenges and stressful environments. These facets make PsyCap a strong personal resource for employees' development. Investigating their role within a crisis context would be of special interest for both employees and organizations at large. Research has demonstrated that PsyCap is important for various work-related outcomes such as organizational commitment, job performance, and employees' well-being [13–15].

PsyCap represents a pivotal and highly valuable personal resource that can facilitate employees' experiences during times of crisis such as with their adaptability. For instance, current research uses resilience to explain the role of organizational adaptation and theories dealing with it, reporting to be the most effective mechanism employees can utilize to deal with crises [16]. In addition, PsyCap facets might facilitate individuals with coping and adapting to their careers [17–19]. For example, factors such as resilience are shown to help employees handle difficult work situations (i.e., by being more persistent), primarily informed by their ability to recover quickly from critical situations [20]. Thus, we expect that PsyCap may directly stimulate employees' career engagement and improve their current environment by affecting their proactive career behavior [21]. Moreover, because PsyCap is reported to help employees with their job-related success, e.g., job commitment [22], we expect that it also enhances employees' subjective career success (i.e., one's subjective evaluation of one's career satisfaction [23]). For example, resilience increases individuals' flexibility and adaptability, especially in uncertain situations [21].

Second, we investigated the mechanisms that could explain why and how PsyCap supports employees with career-related outcomes. PsyCap enhances employees' ability to adapt to new situations [24]; therefore, it may also support their career development (e.g., networking) during the corona crisis or the anticipation of upcoming changes due to COVID-19. To explain the role of PsyCap, we utilized a psychosocial construct, which has been reported to help employees anticipate, prepare, and cope with dynamic and changing work contexts, namely career adaptability (i.e., utilization of skills and personal dispositions to adapt to new circumstances [25]). Composed of vital competencies, behaviors, and attitudes [26], career adaptability may be the key mechanism to facilitate employees as they cope with emerging career concerns [27]. Career adaptability has been posited to yield implications for employees' career satisfaction and their self-rated career performance [9,28] as well as for job performance, especially in demanding environments [29]. Thus, we assume that career adaptability mediates the relationship between PsyCap and the four career-related outcomes. For instance, resilience has been linked directly with employees' flexibility, which eases their adaptation to challenging and uncertain environments [30]. Successfully adapting to the work-related changes due to COVID-19 may result in less career-related worries. Therefore, career adaptability is yet another important construct for employees' careers and we are not aware of any study that has looked into its role related to our chosen career-related outcomes, especially in a crisis context.

Finally, because COVID-19 is a difficult crisis to deal with, we expect that the role of organizational resources will be crucial in facilitating employees' experiences during

this pandemic. Thus, we explored the role of supervisors during the COVID-19 pandemic and added to research on the relevance of leadership for employees' careers. Therefore, we propose that PsyCap may operate to its best when employees have a high-quality relationship with their respective leaders. PsyCap may foster employees' efforts to adapt to new work and career situations; however, efforts may be futile without the trust and support from leaders. Hence, we use leader–member exchange (LMX) as a moderator.

In sum, we propose a moderated mediation research model and expect that the quality of LMX will moderate the relationship between PsyCap and career adaptability. Specifically, we assumed that the potential of employees' PsyCap during the COVID-19 pandemic would be enhanced by the quality of the relationship with their supervisors. To consider the consequences of high career adaptability during the COVID-19 pandemic, we refer to four different career-related outcome variables: career satisfaction, career engagement, career-related COVID-19 worries, and coping with changes in career due to COVID-19. We summarized the relationships between our study variables in Figure 1.

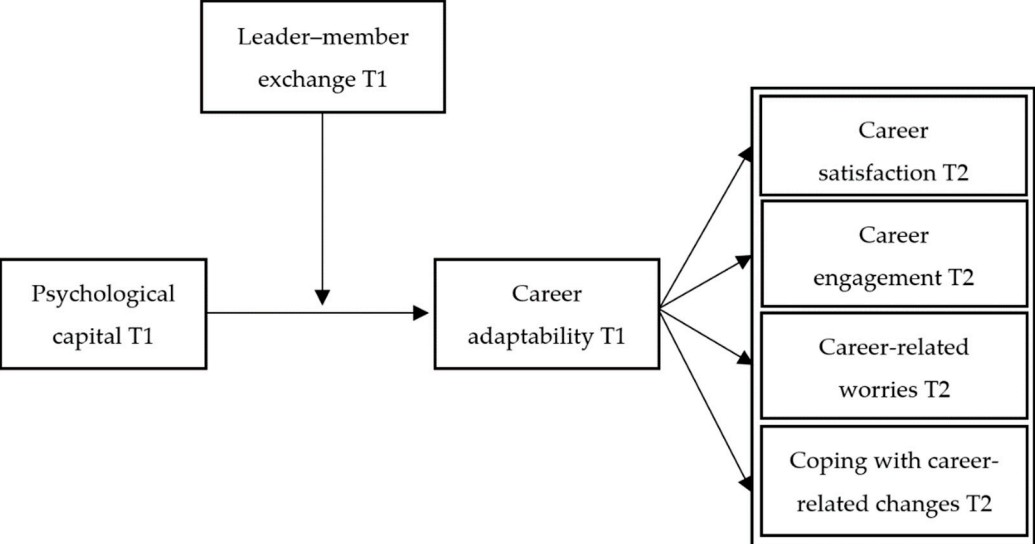

**Figure 1.** Research Model for the Hypothesized Relationships Between Variables.

## 2. Theoretical Background

### 2.1. Psychological Capital

When employees experience stressful situations and rapid environmental changes (e.g., during the COVID-19 pandemic), their PsyCap can act as an important personal resource enhancing their innovative behavior, reducing work-related stress [31], and fostering their well-being (for a review, see Avey et al., [22]). PsyCap refers to individuals' positive psychological resources that facilitate experiences referring to their attitudes, behaviors, and performances [13]. Research shows that the four components of PsyCap (efficacy, hope, optimism, and resilience) facilitate the functioning and well-being of individuals, especially in uncertain and challenging environments [13,22,32].

The first facet of PsyCap, efficacy, is "one's conviction (or confidence) about his or her abilities to mobilize the motivation, cognitive resources, and courses of action needed to successfully execute a specific task within a given context" ([33], p. 66). The second facet, hope, has been defined as "persevering toward goals and, when necessary, redirecting paths to goals (. . . ) in order to succeed" ([22], p. 128). This construct is reported to help individuals with their daily performances [22] and to cope with adverse environments [34]. Third, the optimism component is defined as ". . . an explanatory/attribution style interpreting negative events as external, temporary, and situation specific. . . " ([22], p. 130). Optimism has been shown to help employees with their work, for example, by reducing job stress and conflicts as well as increasing job satisfaction and job performance [35]. The

fourth and final facet of PsyCap, resilience, is described as one's ability to overcome, persevere and bounce back, as well as reach out to gain new knowledge and experiences, and improve relationships with others and find meaning throughout life [36]. Resilient people possess a high level of crucial factors for their development and growth (e.g., self-efficacy, self-awareness), thus this facet can also help with challenging environments [22,37].

From the perspective of Conservation of Resources (COR) theory, each facet of PsyCap could be regarded as a personal resource [38]. COR theory posits that individuals aim to save and gain resources and avoid resource losses, because losing resources elicits stress reactions. According to COR theory [38], resources can boost each other (the authors called this principle a resource "caravan"). In line with this, the authors of PsyCap [39] describe the four components of PsyCap (hope, efficacy, resilience, optimism) as the "HERO within" (p. 180), alluding that they can best facilitate employees' coping with challenging times acting as a single body. Others noted that a caravan of resources is composed of employees' personal resources, which promote and improve functioning [15]). Hence, by acting as one body and a caravan of resources, PsyCap facets may support and strengthen employees' career-related outcomes.

*2.2. Psychological Capital and Career-Related Outcomes*

With our research, we aim to investigate the relationship between PsyCap and career-related outcomes. Career-related outcomes refer to many different constructs and there has been an ongoing discussion concerning: (a) which outcomes are the important ones, and (b) how they can be measured [40–42]. Accounting for the implications of the COVID-19 pandemic for employees' careers [43,44], we aimed at contributing to research by studying the mechanisms related to coping with changes in career due to COVID-19 and COVID-19-related career worries. In addition, to further understand the role of employees' personal resources for relevant career outcomes, we investigated career engagement and career satisfaction as outcomes.

2.2.1. Psychological Capital and Career Satisfaction

Subjective career satisfaction covers individuals' evolvement of their career and relates to the various psychological and work-related outcomes and achievements they gain during work [7]. Subjective career success refers to individuals' occupational life course and the evaluation of personal development and career progress [45]. Subjective career success evaluations are derived from personal perceptions and affective reactions related to their careers such as job success, interpersonal success, hierarchical (i.e., promotion-related) success, life success [46,47], or career satisfaction [46].

Employees have to deal with career hurdles and various obstacles they face during their occupational life while working towards their career goals [46]. According to COR theory, these obstacles can hinder employees' career satisfaction by dissipating their resources, making it more difficult to achieve their career goals. For instance, it is reported that such hurdles could shift employees' attention away from resources (e.g., feedback-seeking), leading to resource depletion and, as a result, to fewer job-related or career-related attainments [46]. Thus, PsyCap should serve as a central resource and help employees through future-focused orientations (i.e., hope and optimism) and help them to thrive in challenging and uncertain environments [13]. Hence, we hypothesize the following:

**Hypothesis 1a:** *PsyCap (T1) will be positively related to career satisfaction (T2).*

2.2.2. Psychological Capital and Career Engagement

Career engagement is a core variable used to explain proactive behaviors and is defined as "the degree to which somebody is proactively developing his or her career as expressed by diverse career behaviors" ([48], p. 577). Proactive personalities are described as individuals who "identify opportunities and act on them; they show initiative, take action, and persevere until they bring about meaningful change" ([21], p. 532). This means

that proactive individuals take deliberate actions to influence their environment [49] by anticipating, planning, and striving actively for impact [50]. Future focus and explicit initiation of actions towards improving their current situations (i.e., employees' proactive behaviors or future temporal focus) are shown to be related to job performance and career outcomes such as adaptability and other organizational outcomes such as team effectiveness [7,20,51,52]. Proactive individuals benefit through their feedback-seeking behaviors, negotiating assignments and roles, or searching and identifying opportunities for their career, for example, by engaging in career planning [53].

Personal resources are critical for, and can foster, employees' proactive career behavior [9,54]. For instance, employees need to invest resources to obtain a better fit with their (occupational) environment and thereby gain further resources [55]. We assume that individuals with high PsyCap (i.e., high hope and optimism) are more likely to act proactively due to their focus on positive outcomes. Hence, we hypothesize the following:

**Hypothesis 1b:** *PsyCap (T1) will be positively related to career engagement (T2).*

2.2.3. Psychological Capital and the COVID-19 Pandemic

Evidence shows that PsyCap may be essential when employees experience resource losses due to challenging environments [56,57]. Recent research reports that individuals high on PsyCap express higher volition and career adaptability [57] as well as significantly lower depression and anxiety and higher satisfaction with life [58]. Therefore, we expected that PsyCap would be a valuable resource, which would help employees deal with the current COVID-19 pandemic. Two of our outcomes directly refer to the connection between employees' careers and the pandemic, namely coping with career-related changes due to COVID-19 and career-related COVID-19 worries.

Coping has been defined as "constantly changing cognitive and behavioral efforts to manage specific external and/or internal demands that are appraised as taxing or exceeding the resources of the person" ([59], p. 141). Career research shows that coping with changes is related to both extrinsic (i.e., salary, job performance) and intrinsic (i.e., organizational commitment) career outcomes [60,61]. High PsyCap may help employees cope with changes, because it enables employees to accept changing working conditions (high resilience). Furthermore, high PsyCap fosters an optimistic and hopeful view of employees' upcoming careers, even under difficult conditions such as COVID-19. Finally, PsyCap supplies employees with the energy to adapt to new situations (e.g., through resilience and efficacy). Therefore, we hypothesize the following:

**Hypothesis 1c:** *PsyCap (T1) will be positively related to coping with changes in career due to COVID-19 (T2).*

Recent findings reported that the current COVID-19 pandemic has had major implications for people's lives by increasing their worries about health and work [62–64]. Worries and other work-related ruminative thoughts threaten employees' mental and physical health, because they interfere with recovery processes [65,66]). However, the PsyCap factors of resilience and optimism may buffer this negative view of the future [60]. Employees with high PsyCap may focus on the positive aspects of the present and the future, therefore will be less worried. Thus, we posit that PsyCap mechanisms can mitigate the negative effects of the COVID-19 pandemic and hypothesize the following:

**Hypothesis 1d:** *PsyCap (T1) will be negatively related to career-related COVID-19 worries (T2).*

*2.3. Career Adaptability as a Mediator between Psychological Capital and Career-Related Outcomes*

To explain the relationship between PsyCap and the four career-related outcomes, we assume that employees' career adaptability is one important mechanism. Career adaptability is defined as one's "readiness to cope with predictable tasks of preparing for and

participating in work role and with the unpredictable adjustments prompted by changes in work and working conditions" ([67], p. 254). Career adaptability is an important psychosocial construct that helps employees anticipate, prepare for, and cope with changing work contexts [67]. Furthermore, career adaptability can predict the employment status of individuals [68], enhance their job performance and well-being in dynamic, changing, and uncertain work situations [39], as well as help individuals cope with emerging career concerns [28].

Studies have reported several social, organizational, and psychological aspects influencing employees' career adaptability. Some authors stressed the role of hope in terms of pathway and agency thinking, stipulating that they enhance one's adaptability by motivating employees to search for and find new pathways towards their employment [34]. Other research has stressed the role of resilience as pivotal for adaptability [69]. For example, resilient individuals are reported to display robust coping mechanisms through expressed self-esteem and self-efficacy as well as self-determination and locus of control [70]. Thus, we expect that PsyCap enhances employees' career adaptability and hypothesize the following:

**Hypothesis 2:** *PsyCap (T1) will be positively related to career adaptability (T1).*

Research has reported on many benefits of employees' career adaptability. For example, some noted that adaptability is directly related to coping with transitions and changing environments [6]. Thus, career adaptability can provide individuals with a robust framework that facilitates their interventions based on their circumstances and needs [71]. Furthermore, adaptability is claimed to be a crucial component for employability [72], related directly to factors such as optimism, internal locus of control, or self-efficacy [6]. Similarly, career adaptability predicts career-related skills such as job search self-efficacy [68] and career commitment [73]. Moreover, career adaptability has been shown to increase individuals' engagement in adaptive and proactive behaviors and reduce anxiety [74,75].

In sum, because adaptability provides individuals with a strong framework for their challenging environments, we assume that career adaptability will mediate the relationships between PsyCap and career-related outcomes (career satisfaction, career engagement, coping with changes in career due to COVID-19, and career-related COVID-19 worries). Following the above findings, we hypothesize:

**Hypothesis 3(a–d):** *Career adaptability (T1) will mediate the positive relationship between psychological capital (T1) and career satisfaction (a; T2), career engagement (b; T2), coping with changes in career due to COVID-19 (c; T2), and career-related COVID-19 worries (d; T2).*

*2.4. Leader–Member Exchange as a Moderator of the Relationship between Psychological Capital and Career Adaptability*

LMX refers to the relationship between supervisor and subordinate as a dyadic social exchange process [76]. High LMX indicates a good relationship between leaders and members of an organization, which yields mutually positive affect, higher levels of respect and trust, and a higher level of information exchange as well as lower job stress [76–78]). However, a low-quality exchange relationship can lead to counterproductive behavior, lower levels of interaction and mutual affect, higher employee turnover, and stress [79–81].

There is also evidence for implications of LMX on employees' work and their performance as well as career-related outcomes [10,82,83]. For example, LMX quality was shown to play an important role in employees' employability [10]. Similarly, others found that LMX positively moderated the relationship between job-related activities (i.e., learning) and employability [83]. In addition, many studies have reported on the benefits that a good relationship between members of an organization can yield during crises [84,85]. For instance, it has been reported that through proximity and trust leaders promote positive interactions with members, which influences employees' personal values and systems and enhances the strength of the relationships between members [84]. This, in turn, has positive implications on other employee behaviors such as easing information transfer or

increasing congruence between leaders and members. Such influences can further enhance employees' adaptability skills. For example, previous research has shown that support from the leader can be an important resource and can boost employees' work motivation and extra role behaviors, which can promote one's career [86,87]. Thus, we expect that LMX quality moderates the relationships between employees' PsyCap and their career adaptability. Therefore, we propose the following hypotheses:

**Hypothesis 4:** *LMX (T1) will moderate the relationship between PsyCap (T1) and career adaptability (T1) so that the relationship is stronger when LMX (T1) is high.*

**Hypothesis 5(a–d):** *LMX will moderate the indirect relationship between PsyCap and career-related outcomes (career satisfaction (a; T2), career engagement (b; T2), coping with changes in career due to COVID-19 (c), and career-related COVID-19 worries (d; T2)) via career adaptability (T1) so that the indirect effects are stronger when LMX (T1) is high.*

### 3. Materials and Methods

*3.1. Procedure*

We conducted a two-wave online study with a time lag of six weeks during the summer and autumn 2021 and recruited our convenience sample through SoSci Panel (https://www.soscipanel.de/, accessed on 10 July 2021). The surveys each took around 15 min to complete and were framed as a study on working in times of COVID-19. Participants were invited to take part in our study via e-mail; they had to be at least 18 years old and work at least 19 h per week. Participation was voluntary and the participants did not receive any financial reward. The local ethics committee (University of Bamberg, dossier number 2021-02/04; 17 April 2021) approved our research.

*3.2. Sample*

In total, we had data from 305 participants. We screened the data for response patterns and careless responding, but could not detect any exceptional cases. Additionally, we conducted outlier analyses and detected one participant in an inter-quartile range larger than three for career adaptability. However, excluding this participant from our analyses did not change the result pattern, therefore we included this participant in our final sample. Furthermore, we conducted logistic regression analyses to check for non-random sampling. The analyses produced insignificant results, which supports the assumption of unsystematic drop-out.

Our participants (60.3% female) were on average 43.78 years old (SD = 11.97), had been working in their current job for 12.74 years (SD = 10.90), and in their current organization for 10.03 years (SD = 9.51). On average, participants worked 38.43 h (SD = 8.22) per week, had been working together with their supervisor for 5.42 years (SD = 5.36), with the majority holding a full-time position (69.9%). Most of the participants did not report a leadership position (65.6%), 16.6% held a low-level or a medium-level leadership position, respectively, and 1.1% a high-level leadership position. Participants were employed in different industries such as engineering, health services, childcare and education, banking, IT, or research and science. The largest group of the participants never worked from home (40.9%), 21.7% worked on average one or two days per week from home, 17.9% worked three or four days from home, and 19.6% worked five or more days from home. Four percent of the participants held a bachelor's degree, 48% a master's degree, and 6% a PhD. The majority did not have children (70.6%), while 17.5% had one child, 10.3% had two children, and 1.6% had three children.

*3.3. Measures*

All items were assessed in German. For scales not available in German, we used the back translation procedure suggested by Brislin [88]. In the first survey, participants reported on their psychological capital, their relationship with their supervisor, career

adaptability, and demographic and job-related variables. In the second survey, we assessed several career outcomes (i.e., coping with career-related changes due to COVID-19, career-related COVID-19 worries, career satisfaction, and career engagement).

### 3.3.1. Psychological Capital

To measure PsyCap, we used the 12-item short version of the Psychological Capital Questionnaire (PCQ) developed by Luthans et al., [13]. The PCQ assesses PsyCap on four dimensions: efficacy, hope, resilience, and optimism. Sample items were: "I feel confident in representing my work area in meetings with management" (efficacy), and "if I should find myself in a jam at work, I could think of many ways to get out of it" (hope). Responses were measured using a 6-point Likert scale ranging from 1 (strongly disagree) to 6 (strongly agree). Cronbach's alpha was 0.88. Mind Garden (www.mindgarden.com, accessed on 10 February 2021) approved instrument usage.

### 3.3.2. Leader–Member Exchange

To measure LMX, we used the German version [89] of the LMX-7 scale by Graen and Uhl-Bien [77]. A sample item was: "I have enough confidence in my supervisor to defend his/her decisions." Participants rated each of the 7 items on a 5-point Likert-type scale with the anchors adapted to the respective item (1 = never/not at all/low/completely disagree/very ineffective to 5 = always/very much/high/completely agree/very effective). Cronbach's alpha was 0.91.

### 3.3.3. Career Adaptability

Career adaptability was measured with the Career Adaptability Scale–Short Form (CAAS-SF [90]). A sample item was: "I have the ability to make decisions by myself." The 12 items were rated between 1 (strongly disagree) and 6 (strongly agree). Cronbach's alpha was 0.92.

### 3.3.4. Coping with Career-Related Changes Due to COVID-19

We adapted five items from the measure developed by Judge et al., [61]. A sample item was: "I think I cope with career-related changes due to COVID-19 better than most of those with whom I work." The scale ranged between 1 (strongly disagree) and 5 (strongly agree). Cronbach's alpha was 0.76.

### 3.3.5. Career-Related COVID-19 Worries

To measure career-related worries due to COVID-19 we used a single item developed by Nitschke [19]. The item was: "How worried are you about losing your job?" We asked respondents to rate their worry on a scale from 0 (not worried at all) to 10 (very much worried).

### 3.3.6. Career Satisfaction

We assessed career satisfaction as an indicator for subjective career success with five items ([91], German version by Abele et al., [45]). A sample item was "Overall, I see my professional career in a very positive light." The scale ranged between 1 (strongly disagree) and 5 (strongly agree). Cronbach's alpha was 0.90.

### 3.3.7. Career Engagement

We measured career engagement as an indicator of proactive career behavior. We used the 9-item measure by Hirschi et al., [53] and asked respondents to answer about their experiences in the past six weeks (e.g., "In the past six weeks, to what extent have you—actively sought to shape your professional future?"). Respondents answered on a Likert scale from 1 (little) to 5 (very much). Cronbach's alpha was 0.90.

## 4. Results

### 4.1. Confirmatory Factor Analysis

Before our hypotheses tests, we conducted several confirmatory factor analyses (CFAs) on our data using the software R, including all constructs of our research model for the respective time points. For our T1 data, we first built a model in which we combined all variables (i.e., PsyCap, LMX, career adaptability) to load on a single factor. Next, we modeled our three variables as three separate factors. In the last step, we included PsyCap and career adaptability as hierarchical factors with their respective dimensions (PsyCap—efficacy, hope, resilience, and optimism; career adaptability—control, confidence, curiosity, and concern). The last model showed the best fit, $\chi^2(423) = 874.025$, CFI = 0.900, TLI = 0.891, RMSEA = 0.065, SRMR = 0.069, AIC = 18,411.139, BIC = 18,669.076 (see Table 1).

**Table 1.** Confirmatory Factor Analyses with Different Factor Solutions for Our Main Variables.

| Model | $\chi^2$ Value | df | p | CFI | TLI | RMSEA Value | 95% CI | p | SRMR | AIC | BIC |
|---|---|---|---|---|---|---|---|---|---|---|---|
| T1 One factor | 2499.099 | 434 | <0.001 | 0.544 | 0.512 | 0.137 | (0.132, 0.142) | <0.001 | 0.132 | 20,014.213 | 20,233.732 |
| T1 Three factors | 1371.918 | 431 | <0.001 | 0.792 | 0.776 | 0.093 | (0.087, 0.098) | <0.001 | 0.075 | 18,893.032 | 19,122.702 |
| T1 Three factors with PsyCap and Career adaptability as hierarchical construct | 874.025 | 423 | <0.001 | 0.900 | 0.891 | 0.065 | (0.059, 0.071) | <0.001 | 0.069 | 18,411.139 | 18,669.076 |
| T2 One factor | 1531.937 | 170 | <0.001 | 0.365 | 0.291 | 0.204 | (0.195, 0.214) | <0.001 | 0.233 | 12,235.762 | 12,366.062 |
| T2 Four factors | 274.842 | 164 | <0.001 | 0.948 | 0.940 | 0.059 | (0.047, 0.071) | <0.001 | 0.078 | 10,990.667 | 11,140.512 |

Note. $N$ = 192–253. Factors T1: PsyCap—psychological capital, leader–member exchange, career adaptability; factors T2: career satisfaction, career-related worries (only one item), coping with career-related changes, career engagement. CFI—comparative fit index; TLI—Tucker–Lewis Index; RMSEA—root mean square error of approximation; CI—confidence interval; SRMR—standardized root mean square residual; AIC—Akaike information criterion; BIC—Bayesian information criterion.

For our T2 data, including coping with career-related changes due to COVID-19, career-related COVID-19 worries, career satisfaction, and career engagement, we followed the same procedure and compared a one-factor solution with a four-factor solution. The four-factor model with a covariance between two items assessing coping with career-related changes due to COVID-19 showed the best fit, $\chi^2(164) = 274.842$, CFI = 0.948, TLI = 0.940, RMSEA = 0.059, SRMR = 0.078, AIC = 10,990.667, BIC = 11,140.512 (see Table 1).

### 4.2. Descriptive Statistics and Hypotheses Testing

Table 2 displays means, standard deviations, and correlations between the study variables.

We tested our moderated mediation model using path analysis in R. This procedure provides the advantage of testing all of our hypotheses in a single model. We used bias-corrected bootstrap estimates (10,000 bootstrap samples) and 95% confidence intervals to examine the effects. We z-standardized all variables prior to our analyses. LMX was entered as the moderator moderating the path from PsyCap to career adaptability. We included covariances between the exogenous variables (i.e., PsyCap, LMX, and the interaction between PsyCap and LMX) in our model, because the data were collected in a single questionnaire. To handle missing values, we used the full information maximum likelihood (FIML) estimation.

**Table 2.** Means (M), Standard Deviations (SD), and Correlations Between the Study and Control Variables.

| | M | SD | 1 | 2 | 3 | 4 | 5 | 6 | 7 | 8 |
|---|---|---|---|---|---|---|---|---|---|---|
| 1. Age | 43.78 | 11.97 | | | | | | | | |
| 2. Gender [a] | 1.61 | 0.49 | −0.22 *** | | | | | | | |
| 3. PsyCap (T1) | 4.74 | 0.75 | 0.11 | −0.05 | | | | | | |
| 4. Career adaptability (T1) | 5.04 | 0.68 | 0.11 | 0.08 | 0.67 *** | | | | | |
| 5. LMX (T1) | 3.56 | 0.82 | 0.04 | 0.02 | 0.34 *** | 0.19 ** | | | | |
| 6. Career satisfaction (T2) | 3.74 | 0.73 | −0.01 | 0.05 | 0.43 *** | 0.42 *** | 0.38 *** | | | |
| 7. Career engagement (T2) | 2.75 | 0.96 | −0.13 | 0.19* | 0.07 | 0.32 *** | −0.04 | 0.09 | | |
| 8. Coping with career-related changes (T2) | 3.05 | 0.80 | −0.09 | 0.06 | 0.39 *** | 0.30 *** | 0.18* | 0.28 *** | 0.28 *** | |
| 9. Career-related worries (T2) | 9.82 | 21.59 | 0.02 | −0.08 | −0.21 *** | −0.20 * | −0.14 | −0.30 ** | −0.01 | −0.14 |

Note. $N$ = 159–265. PsyCap—psychological capital, LMX—leader–member exchange. [a] 1—male, 2— female. * $p$ < 0.05. ** $p$ < 0.01. *** $p$ < 0.001.

Our path model showed a good fit [$\chi^2$(8) = 14.946, $p$ = 0.060, CFI = 0.978, RMSEA = 0.053, SRMR = 0.039, AIC = 5,060.834, BIC = 5,194.765]. In H1a–c, we assumed a positive direct relationship between PsyCap and career satisfaction, career engagement, and coping with career-related changes, respectively. Hypothesis 1d predicted a negative association between PsyCap and career-related worries. As Table 3 shows, PsyCap was significantly associated with career satisfaction [est. = 0.39, SE = 0.12, 95% CI (0.13, 0.60)] and coping with career-related changes [est. = 0.28, SE = 0.12, 95% CI (0.03, 0.51)], but not with career engagement [est. = −0.12, SE = 0.12, 95% CI (−0.39, 0.08)] and career-related worries [est. = −0.21, SE = 0.12, 95% CI (−0.45, 0.03)]. Thus, H1a and H1c were supported, whereas H1b and H1d had to be rejected. In H2, we hypothesized that PsyCap and career adaptability were positively related, which could be supported [est. = 0.69, SE = 0.08, 95% CI (0.54, 0.84)]].

**Table 3.** Results of the Path Analysis Related to the Main Study Variables.

| Variable | Career Adaptability | | Career Satisfaction | | Career Engagement | | Coping with Career-Related Changes | | Career-Related Worries | |
|---|---|---|---|---|---|---|---|---|---|---|
| | Est.(SE) | 95% CI | Est.(SE) | 95% CI | Est.(SE) | 95% CI | Est.(SE) | 95% CI | Est.(SE) | 95% CI |
| PsyCap | 0.69 (0.08) | (0.54, 0.84) | 0.39 (0.12) | (0.13, 0.60) | −0.12 (0.12) | (−0.39, 0.08) | 0.28 (0.12) | (0.03, 0.51) | −0.21 (0.12) | (−0.45, 0.03) |
| LMX | −0.04 (0.04) | (−0.13, 0.04) | | | | | | | | |
| PsyCap × LMX | 0.00 (0.05) | (−0.09, 0.12) | | | | | | | | |
| Career adaptability | | | 0.09 (0.11) | (−0.12, 0.32) | 0.29 (0.11) | (0.08, 0.50) | 0.06 (0.12) | (−0.17, 0.29) | −0.03 (0.12) | (−0.28, 0.21) |
| PsyCap via career adaptability | | | 0.06 (0.08) | (−0.09, 0.22) | 0.20 (0.08) | (0.05, 0.36) | 0.04 (0.08) | (−0.11, 0.20) | −0.02 (0.09) | (−0.20, 0.14) |

Note. $N$ = 305. Est.—Estimate. We used bias-corrected bootstrap estimates (10,000 bootstrap samples). PsyCap—psychological capital; LMX—leader–member exchange.

Hypothesis 3a–d stated indirect associations between PsyCap and the outcomes via career adaptability. The indirect effect could be shown for career engagement, est. = 0.20, SE = 0.08, 95% CI (0.05, 0.36), which supports Hypothesis 3b. However, there were no mediating effects concerning career satisfaction, career-related worries, or coping with career-related changes (see Table 3). Hence, Hypothesis 3a, 3c, and 3d were rejected. According to H4, LMX should moderate the relationship between PsyCap and career adaptability so that this relationship was stronger when LMX was high. However, the data showed no interaction effect between PsyCap and LMX in predicting career adaptability, est. = 0.00, SE = 0.05, 95% CI (−0.09, 0.12). Thus, H4 was rejected. Consequently, the moderated mediation hypotheses (H5a–d) were also not supported (see Table 3). Figure 2 shows the resulting path model (only significant relationships are displayed).

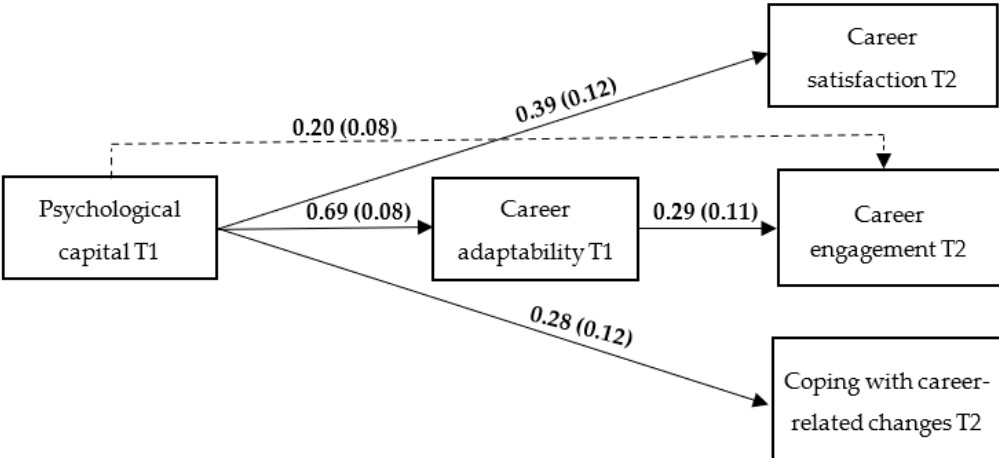

**Figure 2.** Resulting Relationships Between Study Variables. *N* = 305. Only significant paths are displayed. Results show standardized estimates and standard errors in parentheses. The dashed line represents the indirect effect via career adaptability.

## 5. Discussion

The primary objective of this study was to investigate the relationship between PsyCap and four career-related outcomes (career satisfaction, career engagement, coping with changes in career due to COVID-19, and career-related COVID-19 worries). In line with Hypothesis H1a and H1c, our results revealed a positive relationship between PsyCap and career satisfaction and career coping. In addition, we aimed at examining the role of career adaptability as a mediating variable between PsyCap and career-related outcomes. Results revealed a positive relationship between PsyCap and career adaptability, supporting Hypothesis 2. However, career adaptability was only found to be a mediator in the relationship between PsyCap and career engagement. Finally, to examine the implications of social support for employees' careers during the COVID-19 pandemic, we employed LMX as a moderator. Moderation results did not support our expectations, therefore Hypotheses 4 and 5 were rejected.

In line with our results, numerous studies have reported positive correlations between PsyCap and career satisfaction [92]. For example, Al-Ghazali et al., [92] noted that PsyCap can increase employees' career satisfaction through the provision of various positive psychological resources. Thus, similar to these findings and the theoretical assumptions notes (e.g., [38,92]), our study revealed that facets of PsyCap can be essential resources during challenging and uncertain times [92]. By enabling satisfaction with their careers, PsyCap can facilitate employees' careers through the provision of future-oriented focus (i.e., hope and optimism) and by promoting their personal assets (i.e., through efficacy and resilience). For example, hope and optimism might facilitate employees' future focus (i.e., through temporal projections), which would facilitate their future goals and flexibility throughout their career [52].

In addition to career satisfaction, our study revealed that PsyCap could also be an important resource for employees' abilities to cope with shock during crises. These findings are supported by both theoretical claims, (e.g., [38,39]), as well as empirical findings [60]. For instance, Youssef-Morgan and Luthans [38] noted that PsyCap can facilitate employees' coping with challenging times, accounting for the possibility of the four facets acting as a single body. In line with our theoretical assumptions, PsyCap can facilitate employees' well-being by keeping their focus on the positive aspects of the present and the future. These claims are also supported by the COR theory [38], which states the role of saving and gaining resources. Therefore, PsyCap can boost employees' mental and behavioral abilities to cope with stress during uncertain and challenging environments with its facets of hope and resilience.

Nevertheless, our findings did not support our assumptions about the relationship between PsyCap and career engagement and career-related worries. This might indicate that, although satisfied, employees can still struggle to remain engaged in challenging environments. Previous research has reported an unstable relationship between PsyCap and career engagement, stating that this relationship is highly dependent upon different stages of development and career, which can also reflect changing person–environment fit [93]. Thus, the role of PsyCap in this relationship might depend both on the person and their current level of development as well as the environment. Similarly, although PsyCap was shown to be important for coping with changes due to COVID-19, this was not the case with career-related worries. Employees high in PsyCap might have the same worries as employees low in PsyCap, yet employees with high PsyCap reported better coping abilities. This might indicate that individuals might still struggle due to the enhanced stress that worries are shown to yield [65,66]; although, PsyCap can act as an important resource and a good coping mechanism.

Moreover, the positive relationship between PsyCap and career adaptability concurs with previous empirical findings [94]. Research has shown that career adaptability has important implications for employees' abilities to anticipate as well as to prepare for and cope with their career concerns [23] and changing work contexts [67]. For example, Hirschi and Valero [95] found that career adaptability has positive implications for career planning and exploration, as well as negative associations with career decision-making difficulties. In line with these previous findings, our study suggests that PsyCap can further facilitate employees' career adaptability by boosting their adaptability, especially in demanding environments [69]. Thus, our findings improve our understanding of PsyCap for employees' career adaptability in uncertain, stressful, and challenging environments.

We found a partly mediating role of career adaptability between PsyCap and career engagement. In line with theoretical assumptions [67] and previous findings suggesting the significance of career adaptability for employees' proactive behaviors and reduced anxiety [74,75] our findings suggest that PsyCap can boost career adaptability and therefore be beneficial for employees' proactive career behavior (i.e., career engagement). Career-adaptable employees have better coping abilities for dealing with "unpredictable adjustments prompted by changes in work and working conditions" ([67], p. 254). Therefore career adaptability can facilitate employees' careers by proactively seeking social support (e.g., feedback from others), which would in turn enhance their chances for obtaining and maintaining employment [72]. Similarly, because our study focuses on employees working during the COVID-19 pandemic, career adaptability can be especially important for increasing employees' proactive behavior during crises and in challenging environments. Similarly, our findings align with current research reporting on the significance of PsyCap for employees' volition and career adaptability [57] as well as their anxiety and satisfaction [58]. Zhuang et al., [57] reported that employees' low PsyCap can greatly reduce career adaptability, which would in turn have negative implications for their abilities to cope with challenging environments such as the COVID-19 pandemic. Nevertheless, our non-significant findings regarding subjective career satisfaction and career-related COVID-19 worries might indicate the significance of PsyCap even without the role of career adaptability.

Finally, our moderation analysis revealed no significant results concerning the moderating role of LMX between PsyCap and career adaptability. This is yet another critical result and can indicate two important implications. First, in line with various theoretical assumptions [43], our findings suggest that employees high in PsyCap have strong psychological resources that can facilitate their functioning in challenging environments (e.g., adaptability), even when there is a lack of social support. Second, as Avey et al., [96] reported, PsyCap facets are directed primarily towards personal attitudes and emotions, the development and functioning of which might not necessarily depend on their environment. For example, recent findings show that employees' self-beliefs (e.g., believing that abilities are malleable and can be grown through efforts) might moderate employees' attitudes

towards their social environment [97]. In their study, Zyberaj and Volmer [97] found that employees with a growth mindset (i.e., those believing in growth through learning and effort) were willing to accept feedback better than those with a fixed mindset (i.e., rejecting support due to avoidance behavior). In line with these findings, our results suggest that individuals high in PsyCap facets (i.e., efficacy) can improve their adaptability due to their strong beliefs in personal learning and growth. However, those low in PsyCap might view setbacks as permanent failures, which increases chances for avoidance behavior [98]. Therefore, PsyCap can serve employee development and growth in challenging environments by influencing their abilities to cope with challenging contexts [67].

*Practical Implications, Limitations, and Future Research*

The present study yields several implications for both employees and organizations at large. For employees, our study suggests that psychological mechanisms are important for their career development. Thus, whenever possible employees should strive to: (a) enhance their PsyCap, and (b) utilize it during uncertain and challenging times. For organizations, our findings suggest two important implications. First, organizations should consider the role of personality characteristics for employee development and success during their careers. Accounting for the role of PsyCap for several important career-related outcomes in our study, organizations should look for ways to support and enhance this resource, for instance, by providing training related to the development of employees' self-efficacy or resilience. For example, we propose following Bandura's [99] recommendations for strengthening one's self-efficacy as one core facet of PsyCap. According to Bandura [99], there are three effective ways through which one can enhance self-efficacy: (1) through mastery experiences, (2) through vicarious experiences provided by social models, and (3) by social persuasion from others. Aligning with Bandura [99], organizations can utilize role models (i.e., supervisors) to improve employees' efficacy through learning by observation or from supervisors' feedback as a form of persuasion. Similarly, organizations could build employee resilience by increasing employee compassion, shown to enhance resilience. For instance, the compassion cultivation training (CCT) program could be a good training program to increase employee mindfulness and compassion [100]. Employees could also be trained to utilize future-focused orientations, e.g., through temporal projections [51], associated with hope and optimism [101,102].

Second, our findings suggest that, despite the convictions and our expectations on the role of social support (i.e., LMX), it might be important for organizations to provide employees with more autonomy. Research has long reported on the role of autonomy for one's career and well-being [103,104]. For example, Tabiu et al., [104] found that job autonomy significantly influenced employees' adaptive performance, reporting that providing higher job autonomy "will make the employees to be motivated and positively exhibit more positive adaptive behaviors that will enhance organizational effectiveness" (p. 721). Thus, for employees to benefit from social support, organizations should aim at balancing the settings between the provision of social support and autonomy, especially for employees high on PsyCap. Similarly, other research found that employees' proactive behaviors are predictors for employee development. For instance, research on feedback-seeking behavior shows that employees who seek feedback have higher chances for better career success [105]. Thus, in addition to the provision of autonomy for employees, organizations can aim at creating a climate and working culture that stresses the role of feedback-seeking behaviors. In this way, employees can function autonomously and seek help and support whenever they feel the need.

Our research is not without limitations, which points out the way for future research. First, because the sample of this study was German speaking, current findings might not be generalizable to other populations. Therefore, future research should replicate our study with different samples. Second, because we were mainly interested in investigating the implication of PsyCap during the COVID-19 pandemic, results might not be as valid in a normal and no crisis context; therefore, it would be beneficial if current findings were

replicated within a different context. Third, although LMX did not directly moderate the relationship between PsyCap and career adaptability, we encourage future research to replicate our study and utilize additional forms of the LMX measures. For example, future research can use teams' ratings for LMX. This could also help mitigate the common method bias.

## 6. Conclusions

Our findings suggest that psychological capital is an essential resource for employees' careers and well-being, including the contexts of crises and challenging environments; therefore, employees and organizations should aim at fostering and developing it. As a result, employees would enhance their ability to cope with continuously uncertain and challenging work environments.

**Author Contributions:** Conceptualization, J.Z., S.S., A.F.S., L.P., S.R.-K. and J.V.; methodology, J.Z., S.S., A.F.S., L.P. and S.R.-K.; software, A.F.S., S.R.-K. and L.P.; writing—original draft preparation, J.Z., S.S., A.F.S., L.P. and S.R.-K.; writing—review and editing, J.Z., S.S., A.F.S., L.P., S.R.-K. and J.V. All authors have read and agreed to the published version of the manuscript.

**Funding:** This research received no external funding.

**Institutional Review Board Statement:** Our research was approved by the ethics committee of the University of Bamberg (dossier number: 2021-02/04; date: 17 April 2021).

**Informed Consent Statement:** Informed consent was obtained from all subjects involved in the study.

**Data Availability Statement:** The data presented in this study are available on request from the corresponding author.

**Conflicts of Interest:** The authors declare no conflict of interest.

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
