# Peer review of "Developing Sustainable Careers during a Pandemic: The Role of Psychological Capital and Career Adaptability"

_sustainability, doi:10.3390/su14053105_

Round 1

Reviewer 1 Report

This paper addresses the challenging issue of understanding the effects of the pandemic of employees. In its current state I have recommended that this paper is not a good fit for this journal. 

updated: I fundamentally disagree with the individualised focus that the manuscript proposes. There is a place for understanding these constructs but the paper is too deeply positioned at the individual level. The introduction is also very repetitive and does not adequately construct a need for this research. The authors also claim that a longitudinal design has been used. First they claim that two time points are used (which does not qualify a longitudinal design) yet after reading I can only see a procedure that aligns with a cross-sectional approach. 

Author Response

Thank you very much for your time and effort in reviewing our manuscript. We have submitted our manuscript to the special issue of Sustainability entitled “Employability and career success in times of COVID-19”. Employability is defined as the one’s “ability to identify and realize career opportunities”, (Fugate et al., p. 23)[1], where constructs such as career adaptability are reported to play a major role. Thus, although we did not directly investigate employability, we cover important constructs that could be part of the broader concept of employability. Furthermore, we investigate important work-related outcomes, namely career engagement, career satisfaction, career-related COVID-19 worries, and coping with career-related changes due to COVID-91. We also provide several suggestions on improving the PsyCap. One important result is that PsyCap can foster employees’ career adaptability, which is important for managing their employability in effective and sustainable ways. Therefore, with our current findings, we believe that our manuscript is a good fit for this special issue and will contribute to advancing our understanding on the role of personal and social factors for employees’ abilities to attain and maintain employment. This contribution is especially important for the context of crises.

  1. [1] Fugate, M.; Kinicki, A. J.; Ashforth, B. E. Employability: A psycho-social construct, its dimensions, and applications. Vocat. Behav. 2004, 65, 14–38, doi.org/10.1016/j.jvb.2003.10.005.

updated response:

Thank you for your comment. We would like to note a couple of points, which we hope will help clarify your concerns raised.

First, our "individualised focus" is primarily guided by our main predictor, Psychological Capital (PsyCap). Composed of efficacy, hope, optimism, and resilience, PsyCap is considered a personal resource, and therefore, PsyCap is usually investigated at an individual level. Because we were interested in the role of personal resources (i.e., PsyCap) in a crisis context, we decided to focus on an individual level. Furthermore, the focus on individuals might directly reveal how employees could deal with crises. Nevertheless, our focus goes beyond individuals, because we also investigated the role of organizational factors by using leader-member exchange (LMX). On page 3, lines 105-112, we noted: "Finally, because COVID-19 is a difficult crisis to deal with, we expect that the role of organizational resources will be crucial in facilitating employees' experiences during this pandemic. Thus, we explored the role of supervisors during the COVID-19 pandemic and added to research on the relevance of leadership for employees' careers. Therefore, we propose that PsyCap may operate to its best when employees have a high-quality relationship with their respective leaders. PsyCap may foster employees' efforts to adapt to new work and career situations; however, efforts may be futile without the trust and support from leaders. Hence, we use leader-member exchange (LMX) as a moderator." We believe this is another important addition to our research, which goes beyond the individual level.

Second, we have looked into the introduction and revised it in relation to your helpful comment. For example, on page 3, lines 102-104, we added, "This is yet another vital construct for employees' careers, and we are not aware of any study that looked into this role of career adaptability, especially in a crisis context." This and other added parts should add to the clarity in relation to the importance and need for our research.

Finally, with our research design, we aimed at reducing the common method bias. Similar to many papers published by the MDPI (e.g., Meseguer de Pedro et al. 2021; Arku et al. 2022; Falco et al., 2022), we believe that our research with two measurement points does qualify for a longitudinal design (summer and autumn of 2021). In sum, we think that our study contributes to the understanding of the role of PsyCap for employees' careers in a crisis context. We also investigated the role of organizations by shedding light on the moderating role of LMX. Finally, by using two different measurement points, we reduced common method bias, which improved our research design.

References

Meseguer de Pedro, M., Fernández-Valera, M. M., García-Izquierdo, M., & Soler Sánchez, M. I. (2021). Burnout, psychological capital and health during COVID-19 Social Isolation: A longitudinal analysis. International Journal of Environmental Research and Public Health, 18(3). https://doi.org/10.3390/ijerph18031064

Arku, D., Bingham, J. M., Turgeon, J., Michaud, V., Warholak, T., & Axon, D. R. (2022). Exploring the Association of the COVID-19 pandemic on employee sleep quality at a healthcare technology and services organization. COVID, 2(2), 168–174. https://doi.org/10.3390/covid2020012

Falco, A., Girardi, D., Carlo, A. de, Arcucci, E., & Dal Corso, L. (2022). The perceived risk of being infected with COVID-19 at work, communication, and employee health: A longitudinal application of the job demands–resources model. Sustainability, 14(2), 1037. https://doi.org/10.3390/su14021037

Reviewer 2 Report

The study presented in this article approaches a very interesting and timely topic with clear practical implications.

The article is very well structured and written. The research hypothesis are clearly formulated and the statistical analysis is sound. The results are discussed in the context of relevant literature.

I have a few minor suggestions:

L298: “The local ethics committee approved our research”. The authors are affiliated to three different universities, so they should specify which ethics committee approved this research, the number and date of approval.

Table 2: Please replace “Gendera” with “Gender”

L506-507: “Thus, whenever possible, employees should strive to (a) enhance their PsyCap […]”.  The authors should go further and provide the way/s in which the employees could enhance their PsyCap.

Author Response

Point 1: The study presented in this article approaches a very interesting and timely topic with clear practical implications.

The article is very well structured and written. The research hypothesis are clearly formulated and the statistical analysis is sound. The results are discussed in the context of relevant literature.

Response:

Thank you very much for reviewing our manuscript and for your positive feedback. We are happy to hear this.

Point 2: L298: “The local ethics committee approved our research”. The authors are affiliated to three different universities, so they should specify which ethics committee approved this research, the number and date of approval.

Response:

Thank you for pointing that out. The study was approved by the ethics committee of the University of Bamberg (dossier number: 2021-02/04; date: 17.04.2021). We have added information about the ethics committee approval (see lines 323-324).

Point 3: Table 2: Please replace “Gendera” with “Gender”

Response:

Thank you for that comment. The table and the complete manuscript have been thoroughly revised and checked for other typos. A at the end of "Gender" refers to the Gender defined under the Table (see Notes under the Table).

Point 4: L506-507: “Thus, whenever possible, employees should strive to (a) enhance their PsyCap […]”.  The authors should go further and provide the way/s in which the employees could enhance their PsyCap.

Response:

Thank you for pointing that out. We now discuss several additional ways through which organizations can enhance their employee PsyCap. For instance, we recommend that compassion cultivation training (CCT) developed by the University of Stanford. With the current additions, we cover all four facets of PsyCap. Please find our further suggestions between lines 598-603.

Reviewer 3 Report

First, I would like to thank the opportunity to read this very interesting study. This study encloses the analysis of a moderated mediated relationship and has a longitudinal design. I congratulate the authors for having tested such a complex model with a longitudinal design. Moreover, this study was conducted during the COVID-19 pandemic and includes the analysis of the relationships among psychological capital, career adaptability, career-related outcomes and LMX. n addition, the authors used relevant and updated literature to sustain their hypotheses, and the findings and practical implications of the study were deeply discussed. However, the procedures regarding the data analysis need to be clarified.

Below are some comments hoping they contribute to the improvement of the paper.

  • I think there is no need to use the acronym “CA”. I would like to suggest to write always it full: career adaptability. The same with “Subjective career success (SCS)”. In addition, add the meaning of the acronym “COR” the first time you use it (see line 135).
  • Line 156-157: « Career success covers individuals’ evolvement of their career and relates to the various psychological and work-related outcomes and achievements they gain during work (Seibert et al., 1999).» Please amend the reference in line with the references style used in the Sustainability journal. The same regarding line 489: (Pathak & Joshi, 2020).
  • Line 412: Please amend this typo “(tanle 4().”.
  • Regarding the data analysis, since this study has a two-wave design, I would like to suggest clarifying in Figure 1 the time (Time 1 or Time 2) in which each of the variables included in the model was assessed. Also, in the description of the data analysis was not clear for me the use of the longitudinal design. For instance, concerning “Hypothesis 1a: PsyCap will be positively related to subjective career success.”, Psycap was measured at time 1 and subjective career success was measured at time 2?

Author Response

Point  1: First, I would like to thank the opportunity to read this very interesting study. This study encloses the analysis of a moderated mediated relationship and has a longitudinal design. I congratulate the authors for having tested such a complex model with a longitudinal design. Moreover, this study was conducted during the COVID-19 pandemic and includes the analysis of the relationships among psychological capital, career adaptability, career-related outcomes and LMX. In addition, the authors used relevant and updated literature to sustain their hypotheses, and the findings and practical implications of the study were deeply discussed. However, the procedures regarding the data analysis need to be clarified.

Response:

Thank you very much reviewing our manuscript and for the positive feedback.

Point 2: I think there is no need to use the acronym “CA”. I would like to suggest to write always it full: career adaptability. The same with “Subjective career success (SCS)”. In addition, add the meaning of the acronym “COR” the first time you use it (see line 135).

Response:

Thank you for this suggestion. We have amended the acronyms accordingly. In addition, we have added the COR (Conservation of Resources) acronym.

Point 3: Line 156-157: « Career success covers individuals’ evolvement of their career and relates to the various psychological and work-related outcomes and achievements they gain during work (Seibert et al., 1999).» Please amend the reference in line with the references style used in the Sustainability journal. The same regarding line 489: (Pathak & Joshi, 2020).

Response:

Thank you. We have noted the changes according to Sustainability reference style.

Point 4: Line 412: Please amend this typo “(tanle 4().”.

Response:

Thank you. We have amended this mistake. Tables have been thoroughly revised.

Point 5: Regarding the data analysis, since this study has a two-wave design, I would like to suggest clarifying in Figure 1 the time (Time 1 or Time 2) in which each of the variables included in the model was assessed. Also, in the description of the data analysis was not clear for me the use of the longitudinal design. For instance, concerning “Hypothesis 1a: PsyCap will be positively related to subjective career success.”, Psycap was measured at time 1 and subjective career success was measured at time 2?

Response:

Thank you very much for this important notice. This has been a very helpful feedback and has helped us reconsider important points concerning the analyses and reporting of the results. We have followed your recommendation and made changes throughout the manuscript, especially in relation to the results. We started with Figure 1 and added our measurement points (T1 = Time 1 and T2 = Time 2). Similarly, we have further conducted more analyses and amended the overall results in the Result section. Based on our new analyses (i.e., path analysis), we have also added a new Figure (Figure 2), where we introduce paths of our significant results. This way, the reader will be able to easily look into our key findings. Thus, current analyses and description should provide a clear indication of our longitudinal design. For example, in Figure 1, we noted that PsyCap (as a predictor) was measured in T1, while subjective career success (i.e., career satisfaction) in T2. This should provide a clear indication of the hypotheses and the times we measured each of the variables.